# Anticancer Activity of Benzo[*a*]phenoxazine Compounds Promoting Lysosomal Dysfunction

**DOI:** 10.3390/cells13161385

**Published:** 2024-08-20

**Authors:** João Carlos Canossa Ferreira, M. Sameiro T. Gonçalves, Ana Preto, Maria João Sousa

**Affiliations:** 1Centre of Molecular and Environmental Biology (CBMA), Department of Biology, University of Minho, Campus of Gualtar, 4710-057 Braga, Portugal; apreto@bio.uminho.pt (A.P.); mjsousa@bio.uminho.pt (M.J.S.); 2IBS-Institute of Science and Innovation for Bio-Sustainability, University of Minho, Campus of Gualtar, 4710-057 Braga, Portugal; 3Centre of Chemistry (CQUM), Department of Chemistry, University of Minho, Campus of Gualtar, 4710-057 Braga, Portugal; msameiro@quimica.uminho.pt

**Keywords:** Nile Blue analogue, benzo[*a*]phenoxazine, anticancer drug, colorectal cancer, breast cancer, lysosome membrane permeabilization

## Abstract

Specific cancer therapy remains a problem to be solved. Breast and colorectal cancer are among the cancers with the highest prevalence and mortality rates. Although there are some therapeutic options, there are still few effective agents for those cancers, which constitutes a clinical problem that requires further research efforts. Lysosomes play an important role in cancer cells’ survival, and targeting lysosomes has gained increased interest. In recent years, our team has been synthetizing and testing novel benzo[*a*]phenoxazine derivatives, as they have been shown to possess potent pharmacological activities. Here, we investigated the anticancer activity of three of the most potent derivatives from our library, **C9**, **A36**, and **A42**, on colorectal- and breast-cancer-derived cell lines, and compared this with the effect on non-neoplastic cell lines. We observed that the three compounds were selective for the cancer cells, namely the RKO colorectal cancer cell line and the MCF7 breast cancer cell line. In both models, the compounds reduced cell proliferation, cell survival, and cell migration, accumulated on the lysosome, and induced cell death accompanied by lysosomal membrane permeabilization (LMP), increasing the intracellular pH and ROS accumulation. Our results demonstrated that these compounds specifically target lysosomes from cancer cells, making them promising candidates as LMP inducers for cancer therapy.

## 1. Introduction

Globally, the burden of cancer mortality and incidence is rising quickly, to the extent that cancer is now considered to be the leading cause of death and the most important barrier to increasing life expectancy around the world [1]. In the top three most commonly diagnosed cancers, breast cancer ranks first and colorectal cancer ranks third, with breast cancer being the leading cause of cancer-related death in women and colorectal cancer being the second leading cause of cancer-related death in both sexes [2].

Both colorectal and breast cancer are highly heterogeneous diseases with different risk factors, treatment outcomes, and prognoses depending on their molecular subtypes [3,4,5].

The current treatment options for these patients include surgery, radiation, chemotherapy, immunologic therapy, and some targeted therapies, but with limited results. Tumor resistance and severe side effects continue to be major therapeutic limitations, despite recent efforts to develop new therapies leading to improved survival rates. This highlights the need for the discovery of new, effective drugs [6,7,8,9].

In this field, one of the primary drivers in the discovery of drugs for therapeutical applications is the synthesis of novel organic compounds [10,11,12]. Among these, phenoxazine derivatives are of great interest, as they have been reported to possess a variety of pharmacological properties, such as antimicrobial, antifungal, antiviral, and antitumor activities [13,14,15,16,17,18,19]. Within the family of phenoxazine derivatives, benzo[*a*]phenoxazines have gained importance because, in addition to their fluorescent properties, they have shown potent antitumor activity [20,21,22,23]. However, information about their targets and mechanism of action is still very limited.

Our research team has been focused on synthesizing new benzo[*a*]phenoxazine derivatives, evaluating their pharmacological activity, and elucidating their mechanisms of action in order to assess their potential applications. To classify the most potent synthetized compounds, we applied a comparative approach involving an evaluation of their biological activity by determining their MIC (minimum inhibitory concentration), using a standard microdilution method for activity testing with the yeast *S. cerevisiae* as a eukaryotic cell model [17,24,25,26,27,28,29,30,31,32,33,34,35,36,37,38]. We further evaluated one of our compounds, BaP1 (a lead compound displaying an MIC of 25 μM), against colorectal cancer cells, and discovered that it had promising anticancer activity. We discovered that this effect was directly related to its lysosomal accumulation and lysosomal membrane permeabilization [39].

In the present work, we evaluate the anticancer potential of the most potent compounds from our library, **C9**, **A36**, and **A42** with MICs of 6.25 μM, 1.56 μM, and 0.78 μM, respectively (Figure 1) [31,40,41,42,43]. Our aim is to uncover their anticancer activity not only against colorectal cancer, but also test their potential against breast cancer cells, as well as to evaluate their intracellular targets.

## 2. Materials and Methods

### 2.1. Compounds, Cell Lines, and Culture Conditions

The compounds **C9**, **A36**, and **A42** were synthesized by our research team according to previously published experimental procedures [25,40,41,42,43]. ^1^H NMR spectra were used to confirm the structure and purity of each compound. It was verified that all compounds were >95% pure by ^1^H NMR. 

We used two non-neoplastic cell lines, NCM460 cells derived from healthy mucosal epithelium and BJ-5ta fibroblast cells isolated from the foreskin of a male patient, obtained from InCell, San Antonio, TX, USA and ATCC (American Type Culture Collection), Manassas, VA, USA respectively. We also used three CRC-derived cell lines, RKO, SW480, and HCT116, and two breast-cancer-derived cell lines, MDA-MB-231 and MCF7, all obtained from ATCC. 

The NCM460 and SW480 cells were grown in Roswell Park Memorial Institute (RPMI) 1640 media containing stable glutamine (Biowest, Nuaillé, France), 1% penicillin–streptomycin (Biowest), and 10% heat-inactivated fetal bovine serum (FBS; Gibco, Invitrogen, Carlsbad, CA, USA). The HCT116 cells were grown in McCoy’s 5A Medium (Biowest) containing 1% penicillin–streptomycin and 10% heat-inactivated FBS (Gibco, Invitrogen). The RKO, MDA-MB-231, and MCF7 cells were grown in Dulbecco’s Modified Eagle’s Medium (DMEM) High Glucose (Biowest), which was supplemented with 1% penicillin–streptomycin (Biowest) and 10% heat-inactivated FBS. The BJ-5ta cells were grown in a 4:1 combination of DMEM (Biowest) and Medium 199 (Biowest), supplemented with 0.01 mg/mL of hygromycin (Sigma, Burlington, MA, USA) and 10% heat-inactivated FBS (Gibco, Invitrogen). The cell lines were plated onto 25 or 75 cm^3^ tissue culture flasks and maintained in a humidified incubator with 5% CO_2_ at 37 °C.

### 2.2. Sulforhodamine B Assay

To calculate their IC_50_ values, the various cell lines were plated into 24-well plates at a density sufficient to achieve 80% confluence after 48 h of growth under normal conditions (no treatment, in the respective culture medium) as follows: 4 × 10^4^ cells/mL (RKO and MDA-MB-231), 5 × 10^4^ cells/mL (HCT116), 6 × 10^4^ cells/mL (MCF7), 8 × 10^4^ cells/mL (BJ-5ta), 1.75 × 10^5^ cells/mL (SW480), and 3 × 10^5^ cells/mL (NCM460). The cells were allowed to adhere overnight, from 12 h to 18 h, and then treated with increasing concentrations of each compound, **C9**, **A36**, and **A42** (0.125 μM, 0.25 μM, 0.5 μM, 1 μM, 2.5 μM, and 5 μM), in their respective culture mediums for 48 h. In addition to a negative control that consisted only of cells and growth medium, a negative control containing the highest concentration of the solvent in which the compound was dissolved (DMSO 0.1%) was also used. The SRB assay was performed according to Marques et al. [44]. The IC_50_ values were obtained using GraphPad Prism version 8.2.1 through the application of a sigmoidal dose–response nonlinear regression, after logarithmic transformation. The selectivity index was determined by dividing the IC_50_ value of the non-neoplastic BJ-5ta cell line by the IC_50_ value of the cancer cell lines (IC_50_ BJ-5ta/IC_50_ cancer cell line); furthermore, for the CRC model, a second selectivity index was also determined using the non-neoplastic NCM460 cell line (IC_50_ NCM460/IC_50_ CRC cell line). According to the results of the selectivity index, the cell line with the highest selectivity value for each of the models, CRC and breast, was selected for subsequent characterization.

### 2.3. Cell Proliferation by CFSE

RKO and MCF7 cell proliferation was evaluated through flow cytometry using the Carboxyfluorescein Diacetate Succinimidyl Ester (CFSE) probe. Confluent cells were collected, rinsed with PBS 1×, and incubated with 5 μM of CFSE for 15 min at 37 °C. Then, the cells were rinsed, resuspended in growth medium, and seeded in 12-well plates at 1 × 10^5^ cells/mL. After 24 h, adherent cells were treated with compounds **C9**, **A36**, and **A42** at IC_50_ and 2 × IC_50_ concentrations or 0.1% DMSO (vehicle). Untreated cells were used as controls. The cells were harvested at different time points of 0 h, 24 h, 28 h, and 72 h. The cells were then analyzed in a flow cytometer and the CFSE median fluorescence intensity was quantified using the FITC-A channel. All median values were normalized to the time point 0 h.

### 2.4. Colony Formation Assay

A colony formation assay was conducted on the RKO and MCF7 cell lines. In 6-well plates, 600 cells/mL and 800 cells/mL RKO and MCF7 cells, respectively, were seeded. Adherent cells were treated with the compounds **C9**, **A36**, and **A42** at IC_50_/2 and IC_50_ concentrations or 0.1% DMSO (vehicle). As a control, untreated cells were used. Following a 48 h incubation period, the cells were twice rinsed with PBS 1× before being supplemented with new medium. After that, the cells were given 10–14 days to grow (new medium was added every 3 days). Following a PBS 1× rinse, the colonies were fixed for 30 min using 0.5% crystal violet and 6% glutaraldehyde. The ImageJ software (V1.53) was used to calculate the number of colonies. The percentage of colonies was determined and normalized for the untreated control. 

### 2.5. Wound Healing Assessment Assay

RKO and MCF7 cells were seeded in 12-well plates at densities of 5 × 10^5^ cells/mL and 4.5 × 10^5^ cells/mL, respectively. After 24 h, a wound was created by scraping the cell layer using a 1 mm pipette tip. The cells were treated with **C9**, **A36**, and **A42** at IC_50_ and 2 × IC_50_ concentrations or 0.1% DMSO (vehicle). Untreated cells were used as controls. The wound regions were photographed at 0 and 12 h, and migration was assessed using the Image J software. The relative migration was calculated in relation to the untreated controls. 

### 2.6. Annexin V/PI Assay

RKO and MCF7 cells were seeded in 12-well plates at a density of 1 × 10^5^ cells/mL. After 24 h, the cells were treated with **C9**, **A36**, and **A42** at 2 × IC_50_ and 4 × IC_50_ concentrations or 0.1% DMSO (vehicle). Untreated cells were used as controls. After 48 h, floating and adherent cells were collected. The cells were resuspended in 100 mL of binding buffer and treated for 15 min in the dark with 5 μL of Annexin V-FITC (Detection Kit-ab14085) and 5 μL of Propidium Iodide (50 μg/mL). The samples were subjected to flow cytometry analysis, PI fluorescence was detected using the ECD-A channel, and Annexin V fluorescence was detected using the FITC-A channel.

### 2.7. Evaluation of ROS Levels

The RKO and MCF7 ROS levels were evaluated through flow cytometry using the dihydroethidium (DHE) probe. RKO and MCF7 cells were seeded in 12-well plates at a density of 1 × 10^5^ cells/mL. After 24 h, the cells were treated with the compounds **C9**, **A36**, and **A42** at 2 × IC_50_ and 4 × IC_50_ concentrations or 0.1% DMSO (vehicle). As a control, untreated cells were used. The cells were collected after 48 h of treatment, washed with PBS 1×, and stained for 30 min at 37 °C in the dark using 0.5 μM DHE. The DHE mean fluorescence intensity was analyzed by flow cytometry using the PE-A channel.

### 2.8. Fluorescence Staining Evaluation

For fluorescence staining observations, RKO and MCF7 cells were co-stained with the compounds and with LysoSensor Green DND-189 and DAPI. The RKO and MCF7 cells were plated on microscopy slides with a density of 1.5 × 10^5^ cells/mL and allowed to adhere for 24 h. On the next day, the cells were exposed to **C9**, **A36**, and **A42** IC_50_ for 3 h. The cells were washed and resuspended in PBS 1× and then stained for 30 min at 37 °C with 2 μM of LysoSensor Green. The cells stained with LysoSensor Green were co-stained with DAPI at a final concentration of 10 μg/mL. The Olympus BX6F2 microscope (40× and 60× oil immersion objectives) (Tokyo, Japan) was utilized to analyze the samples. It was fitted with the appropriate filter cubes, which included U-FDICT (differential interference contrast), TLV-U-FF-FITC (green), U-FYW (far-red), and U-FUNA (blue).

### 2.9. Lysosomal Membrane Permeabilization Assessment

RKO and MCF7 lysosomal membrane permeabilization (LMP) was evaluated through flow cytometry using the Acridine Orange (A.O) probe. RKO and MCF7 cells were seeded in 12-well plates at a density of 1 × 10^5^ cells/mL. After 24 h, the cells were treated with the compounds **C9**, **A36**, and **A42** at 2 × IC_50_ and 4 × IC_50_ concentrations or 0.1% DMSO (vehicle). As a control, untreated cells were employed. Following a 48 h period, the adherent and floating cells were collected and subjected to a 15-min dark incubation period at 37 °C with 1 μM A.O. After that, the samples were examined by flow cytometry using the PC5.5 channel.

### 2.10. Intracellular pH Evaluation

Evaluations of the intracellular pHs (pHi) of the RKO and MCF7 cells were performed through a flow cytometry optimized protocol using the pH-sensitive probe BCECF-AM. RKO and MCF7 cells were seeded in 12-well plates at a density of 1 × 10^5^ cells/mL. After 24 h, the cells were treated with the compounds **C9**, **A36**, and **A42** at 2 × IC_50_ and 4 × IC_50_ concentrations or 0.1% DMSO (vehicle). As a control, untreated cells were used. Attached and floating cells were collected, washed, and then resuspended in Hank’s balanced salt solution (HBSS) after a 48 h treatment period. After that, the cells were stained with 1 μM of BCECF-AM for 30 min at 37 °C. The samples were analyzed by flow cytometry. The FITC-A and PE-A channels were used to detect the fluorescence mean of BCECF-AM. The percentage of cells exhibiting intracellular acidification was estimated from the percentage of cells displaying an FITC-A/PE-A ratio lower than the control cells.

### 2.11. Fluorescence Microscopy and Flow Cytometry General Considerations

An Olympus BX6F2 microscope equipped with filter cubes U-FDICT (differential interference contrast), U-FYW (far-red), U-FGNA (red), and U-FUNA (blue) together with a 40× and 60× oil immersion objective was used for examining the microscopy samples. The Olympus ImageLS software (V2.2) was used to process the photos.

A Beckman Coulter Cytoflex System B4-R2-V0 flow cytometer fitted with a 488 nm solid-state laser (50 mW), FS, SS, FITC (525/40 BP), PE (585/42), ECD (610/20 BP), and PC5.5 (690/50 BP) channels was used to evaluate the flow cytometry samples. Each sample was examined, and twenty thousand cells, at a low flow rate, were considered. The flow cytometric analysis was carried out using the CytExpert Data software (V2.6). 

### 2.12. Data Analysis

Results from at least three independent experiments are presented as means ± SD. Dunnett’s post-test was used in conjunction with one-way or two-way ANOVA to assess the data. *p*-values < 0.05 were considered to be statistically significant. Statistical analyses were carried out on macOS using GraphPad Prism 8.2.1.

## 3. Results and Discussion

### 3.1. Biological Activity of the Benzo[a]phenoxazine Compounds ***C9***, ***A36***, and ***A42***

#### 3.1.1. Effect of **C9**, **A36**, and **A42** on Cell Viability

Three CRC-derived cell lines, SW480 (KRAS^G12V^), HCT116 (KRAS^G13D^), and RKO (BRAF^V600E^), harboring somatic mutations on KRAS or BRAF, and two breast-cancer-derived cell lines, MDA-MB-231 (basal like) and MCF7 (luminal A subtype), together with two non-neoplastic cell lines, NCM460 (normal colon mucosal epithelial cell line) and BJ-5ta (normal foreskin fibroblast cell line), were used to study the activity and selectivity of the three new benzo[*a*]phenoxazine compounds, **C9**, **A36**, and **A42**. All cell lines were treated with a range of drug concentrations (0.125 μM, 0.25 μM, 0.5 μM, 1 μM, 2.5 μM, and 5 μM) for 48 h, and the effect on cell viability was assessed using a sulforhodamine B assay (SRB) (Figure 2). The results were used to determine the IC_50_ value of each compound for all the cell lines (Table 1). In addition, the selectivity index was determined using the BJ-5ta cell line as a reference, as well as the NCM460 cell line for the CRC model (Table 1).

The two non-neoplastic cell lines, BJ-5ta and NCM460, were the most resistant to the compounds and had the highest IC_50_ values (Figure 2 and Table 1). All CRC cell lines were quite sensitive to all compounds. Treatment significantly reduced cell viability, resulting in low IC_50_ values. In particular, the RKO cell line was the most sensitive for the CRC model, showing very low IC_50_ values and very high selectivity indices for all compounds (Figure 2 and Table 1). This is consistent with our previous results, where we tested the anticancer potential of a related benzo[*a*]phenoxazine compound (BaP1–lead compound) and observed that RKO cells were highly sensitive to the drug (Table 1) [39]. This is particularly important, as most colorectal cancer patients displaying the BRAF (V600E) mutational activation present in the RKO cell line display resistance to chemotherapy and targeted therapies such as EGFR inhibitors [45,46,47,48]. 

In the breast cancer model, all three compounds significantly decreased the viability of MCF7 and MDA-MB-231 cells (Figure 2). The compounds **C9**, **A36**, and **A42** showed lower IC_50_ values and a higher selectivity index for the MCF7 cells compared to the MDA-MB-231 cells, indicating a greater effectiveness against the luminal A subtype of breast cancer. Although patients with this subtype may respond to targeted endocrine therapy, such as selective small molecules of estrogen biosynthesis and estrogen receptor modulators, long-term endocrine therapy has been linked to systemic toxicity, tumor resistance, and the emergence of drug-resistant cancer stem cells that interfere with therapy and promote disease progression, emphasizing the need for new chemotherapeutic agents [49,50,51,52].

Overall, all the new tested compounds revealed a high anticancer potential and exhibited lower IC_50_ values and higher selectivity indices than our lead compound BaP1. In particular, the CRC-derived cell line RKO and the breast-cancer-derived cell line MCF7 were found to be the most sensitive cell lines to the three compounds. To better assess the phenotypic alterations associated with the anticancer potential of the compounds and their mechanisms of action, we selected these cell lines and performed a more detailed characterization of the biological activity of the three compounds.

#### 3.1.2. **C9**, **A36**, and **A42** Decrease RKO and MCF7 Cell Proliferation

The benzo[*a*]phenoxazines **C9**, **A36**, and **A42** were tested for their capacity to decrease cell growth using an optimized flow cytometry Carboxyfluorescein Diacetate Succinimidyl Ester (CFSE) labeling procedure. This procedure enables the differentiation of successive rounds of cell division, since the CFSE probe is uniformly distributed among daughter cells during cell division. As a result, a decrease in green CFSE fluorescence was employed as an indicator of cell proliferation.

RKO and MCF7 cells were treated with IC_50_ and 2 × IC_50_ concentrations of the compounds, and the effect on cell proliferation was analyzed after 24 h, 48 h, and 72 h of treatment by examining the decrease in green CFSE fluoresce in comparison with the untreated controls. The three compounds showed a significant effect, as a reduction in the proliferation of the treated cells was observed for the two cell lines, with the highest effect occurring with the compound **A36** (Figure 3A,B). However, this inhibition proved to be much stronger in the breast cancer line MCF7. With the compound **A36**, it was possible to observe that the CFSE cell fluorescence decreased much less in the treated cells than in the untreated controls. We observed that the CFSE median fluorescence of the untreated cells decreased to 37% at 24 h, 21% at 48 h, and 10% at 72 h, in contrast to the 53% IC_50_, 62% 2 × IC_50_ at 24 h, 50% IC_50_, 59% 2 × IC_50_ at 48 h, and 41% IC_50_, 47% 2 × IC_50_ at 72 h (Figure 3B), reaching statistical significance. These results led us to conclude that these compounds are potent proliferation inhibitors, what is an important feature in anticancer drugs.

#### 3.1.3. **C9**, **A36**, and **A42** Reduce Cell Migration of RKO and MCF7 Cells

The development of metastases is one of the most lethal aspects of cancer. Cell migration is a central point in the pathological and physiological development of this process [53]. The wound healing assay is a simple in vitro prediction approach that can be used to assess the inhibitory potency of anticancer drugs against cell migration [54]. To understand whether some of the benzo[*a*]phenoxazine compounds have the ability to inhibit cell migration, we performed a wound healing assay by exposing RKO and MCF7 cells to the IC_50_ and 2 × IC_50_ of the compounds for 12 h. After incubation, wound closure was assessed and the wounds treated with the compounds were compared with the untreated controls (Figure 4). Wound closure in the controls was approximately 20% and 12.5% for the RKO and MCF7 cells, respectively. In the RKO cell line, all compounds had a similar effect at both IC_50_ and 2 × IC_50_, decreasing cell migration by half to values around 10%. A similar decrease was observed in the MCF7 cell line, but 2 × IC_50_ in general, and the compound **C9** in particular, had a more pronounced effect (Figure 4). These results clearly show that all compounds inhibited cell motility, suggesting that they may be able to prevent metastatic dissemination.

#### 3.1.4. **C9**, **A36**, and **A42** Reduce RKO and MCF7 Cell Survival

One of the biggest problems found in chemotherapy regimens is that, even after several cycles, some cancer cells have the ability to resist and relapse, maintaining their malignant behavior [55,56]. Thus, the perfect scenario is to use agents that are effective and prevent/reduce this relapse rate after chemotherapy cycles. With this in mind, we examined the effects of the agents on the survival of the RKO and MCF7 cells by evaluating their effects on the clonogenic ability of the cells (the ability of a single cell to survive and form a large colony), performing a colony formation assay. This assay can be used to determine a cell’s ability to survive exposure to an exogenous drug and form colonies when the substance is removed, replicating in vitro what happens during chemotherapy rounds.

As such, we assessed the effects of **C9**, **A36**, and **A42** at IC_50_/2 and IC_50_ after 48 h of treatment. We found that, in both cell lines, all compounds were able to significantly reduce clonogenic ability (Figure 5). In the RKO cell line, the compound **A42** was the most effective, with reductions of more than 75% and 90% of the colonies formed for IC_50_/2 and IC_50_, respectively. In the breast cancer cell line, these effects were even more pronounced. In fact, we only detected a reduced number of small colonies in the **A36** and **A42** IC_50_/2 conditions (Figure 5A,B).

These results show the efficiency of these compounds in inducing a loss of cellular viability and, thus, reducing colony formation even at low doses, demonstrating their inhibitory effect on cell survival. Cell survival is a balance between proliferation and death, so we next studied the effects of these compounds on cell death.

#### 3.1.5. **C9**, **A36**, and **A42** Induce RKO and MCF7 Cell Death

One of the main goals of clinical oncology is the development of compounds and treatments capable of selectively eliminating cancer cells. In this sense, we next evaluated the capability of the compounds to induce cell death in both cell lines. For this, we performed a flow cytometry analysis using annexin V-FITC and propidium iodide PI double staining. The conjugation of these two probes allowed for the discrimination of viable (negative for both), early apoptotic (annexin + and PI−), late apoptotic (annexin+ and PI+), and necrotic (annexin− and PI+) cells. RKO and MCF7 cells were exposed to 2 × IC_50_ and 4 × IC_50_ of the compounds for 48 h. Both flouting and attached cells were harvested and double stained. The results are summarized in Figure 6. The analysis showed that, globally, all the compounds induced cell death in the RKO and MCF7 cells and mainly increased the number of apoptotic cells (early and late) (Figure 6B). **C9** and **A42** were revealed to be the most powerful compounds, and the MCF7 cell line was the most sensitive cell line. 

Overall, these results show that, at concentrations above IC_50_, these compounds were capable of inducing apoptotic cell death, similar to what was previously observed for our lead compound (BaP1) [39]. This is particularly important, since both RKO and MCF7 cells are known to be intrinsically resistant to apoptotic induction [57,58]. Indeed, our results are consistent with other reports in the literature, where it was shown that other phenoxazine compounds are capable of inducing apoptotic cell death in various cell types [59,60,61,62].

### 3.2. Intracellular Target Evaluation of the Benzo[a]phenoxazine Compounds ***C9***, ***A36***, and ***A42***

#### 3.2.1. **C9**, **A36**, and **A42** Accumulate and Emit Fluorescence on RKO and MCF7 Cell Lysosomes

As mentioned above, our recent work has focused on the synthesis and characterization of new benzo[*a*]phenoxazine compounds. We attempt to preserve the central structures of the molecules and introduce various functional groups to functionalize the compounds. However, we have observed that all these compounds tend to accumulate and emit fluorescence at the vacuolar membrane (yeast) and the lysosomes (human CRC cells) [25,26,39,63]. 

To determine whether this accumulation and emission also occurs for **C9**, **A36**, and **A42**, we took advantage of their intrinsic far-red fluorescence and co-stained RKO and MCF7 cells with the compounds, LysoSensor Green DND-189 (a probe that accumulates and stains lysosomes), and DAPI (stains the nuclei). Using the IC_50_ concentrations of the compounds, we observed that in both the RKO and MCF7 cells (Figure 7A,B), **C9**, **A36**, and **A42** far-red fluorescence appeared in punctuated structures that co-localized with the LysoSensor Green lysosome staining, proving that there was an accumulation of these compounds in the lysosomes.

These observations are consistent with those previously made for this family of compounds, including for the lead compound (BaP1), and with other reports from the literature, showing that compounds of this family accumulate and emit fluorescence in acidic lysosomes [64,65,66].

#### 3.2.2. **C9**, **A36**, and **A42** Lysosomal Membrane Permeabilization, Cytosolic Acidification, and ROS Generation

In our previous observations using BaP1, we found that its accumulation and fluorescence emission at the lysosome (CRC model) or vacuolar membrane (yeast model) were not irrelevant. Indeed, using yeast and colorectal cancer cell lines as complementary models, we observed that these organelles were direct targets of the drug, as we found permeabilization of their membranes and the involvement of vacuolar/lysosomal proteins in BaP1’s toxic effect [26,39].

Following the information obtained with the lead compound and considering the microscopic observations made with these new compounds, we next investigated whether their toxicity was also associated with lysosome targeting. To this end, we evaluated the induction of lysosomal membrane permeabilization (LMP) by flow cytometry, using acridine orange (AO) as probe. This probe allows for an assessment of permeabilization, as its fluorescence decreases upon the release of lysosomal protons. RKO and MCF7 cells were treated with 2 × IC_50_ and 4 × IC_50_ (cell-death-inducing concentrations) of the compounds for 48 h, and were then incubated with AO and evaluated in the flow cytometer. The cells that presented decreased AO fluorescence were considered to have permeabilized lysosomes. As observed in the histograms in Figure 8A, treatment with the compounds resulted in a decrease in AO fluorescence, representing the existence of cell populations with permeabilized lysosomes.

After quantifying the percentage of cells with LMP (Figure 8B), we found that compound **C9** had the greatest effect on both cell lines, with 4 × IC_50_ leading to a very high percentage of cells with permeabilized lysosomes, more than 80% of the total population. These levels are consistent with what we observed for the cell death induction, as the compound **C9** also proved to be the compound with the greatest ability to induce cell death in both cell lines. Moreover, in the case of the compounds **A36** and **A42**, the levels of inviable cells and the levels of cells with LMP were also identical. These correlations suggest that the induction of LMP by these compounds is implicated in their mechanisms of cell death induction.

To further support LMP’s role in cell death induction, we investigated the effects of the compounds on intracellular pH (pHi), as LMP induction is associated with an increase in cytosolic proton levels and, as a result, a reduction in intracellular pH [67,68,69]. We used the dual-emission ratiometric pH-sensitive probe BCECF-AM, which emits fluorescence based on pHi. The result of acidification was shown by a decrease in the fluorescence ratio between BCECF-AM green (FITC-FL1) and red fluorescence (PE-FL2) (FL1/FL2). The cells with lower pHi values could be identified by a lower FL1/FL2 ratio compared to the control cells (C–). This is defined by the gates in the histograms in Figure 8B. Indeed, our results are consistent with LMP induction, as we observed the appearance of populations with lower FL1/FL2 ratios, representing cells with lower pHi values, after 48 h of treatment with the compounds for both cell lines (Figure 8C). Moreover, the compound **C9** showed the most pronounced effect, as expected, demonstrating a direct association between LMP and intracellular acidification.

Since ROS generation is reported to precede LMP and cell death [70], and in view of reports showing that some compounds of this family increase intracellular ROS levels under certain circumstances [71,72], we also investigated whether **C9**, **A36**, and **A42** had any effects in this regard. Indeed, by flow cytometry using dihydroethidium (DHE), a probe that allows for the direct detection of superoxide and hydrogen peroxide, we observed that all compounds, including **C9** with a higher intensity, led to a very significant increase in the ROS levels in the two cell lines, as shown by an increase in the fluorescence mean of DHE (Figure 8D). In fact, these high levels of ROS seem to suggest that there may be a link with LMP, as they may be related to lysosomal membrane damage and consequent lysosomal permeabilization, as occurs in several cases of LMP-associated cell death [73,74,75].

There are several pharmacological studies in the literature that have explored the applications and endogenous targets of phenoxazine derivatives in general [13,14,15,16,17,18,19]. However, as far as benzo[*a*]phenoxazines are concerned, although there are studies demonstrating their suitability as anticancer drugs [20,21,22,23], there is not much information on their mechanisms of action. Our results are relevant, as they contribute to the elucidation of the mechanisms of action of these compounds and show that, at least in the specific case of these compounds, the lysosome arises as their target, which is consistent with what has been previously reported for BaP1 [39].

## 4. Conclusions

Here, we investigated the anticancer activity of the three of the most potent benzo[*a*]phenoxazine derivatives from our library of compounds, **C9**, **A36**, and **A42**, on colorectal and breast cancer cell lines. The three compounds proved to be selective for the cancer cells, improving selectivity compared with our lead compound BaP1. They showed IC_50_ values in the low micromolar range and exhibited greater selectivity indices for the cancer cells, especially for the RKO colorectal cancer cells harboring BRAF^V600E^ mutations and the MCF7 breast cancer cells of the luminal subtype A. We performed a detailed characterization of the biological activity of these compounds and found that they were able to inhibit cell proliferation, cell migration, and cell survival in both cancer models. 

We found that they accumulated in the lysosomes of the RKO and MCF7 cells, and this might be the reason why they were able to induce apoptotic cell death. At cell-death-inducing concentrations, we observed the permeabilization of lysosomal membranes accompanied by intracellular acidification and an increase in intracellular ROS accumulation. These results are consistent with what we observed previously for our lead benzo[*a*]phenoxazine compound BaP1, suggesting that the lysosome is the target of these compounds, but now we extend these results to more active compounds and also to breast-cancer-derived cell lines. Lysosomes are very attractive from a therapeutic point of view, as cancer cells have more numerous, larger, and less stable lysosomes and alterations in their lysosomal structure and function that make them more susceptible to lysosome destabilization. This could be the reason behind the selectivity and higher toxicity of **C9**, **A36**, and **A42** towards cancer cells.

Our work is relevant, as it contributes to the characterization of the mechanisms of action of this family of compounds. Moreover, we show that the anticancer activity and mechanisms of action of these compounds are conserved in different cell tissues, making **C9**, **A36**, and **A42** promising candidates as LMP inducers for cancer therapy.

## Figures and Tables

**Figure 1 cells-13-01385-f001:**
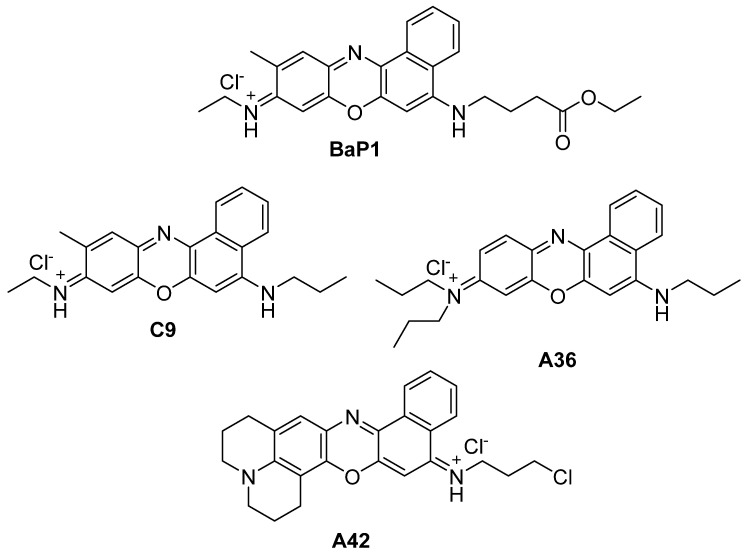
Chemical structures of BaP1, **C9**, **A36**, and **A42**.

**Figure 2 cells-13-01385-f002:**
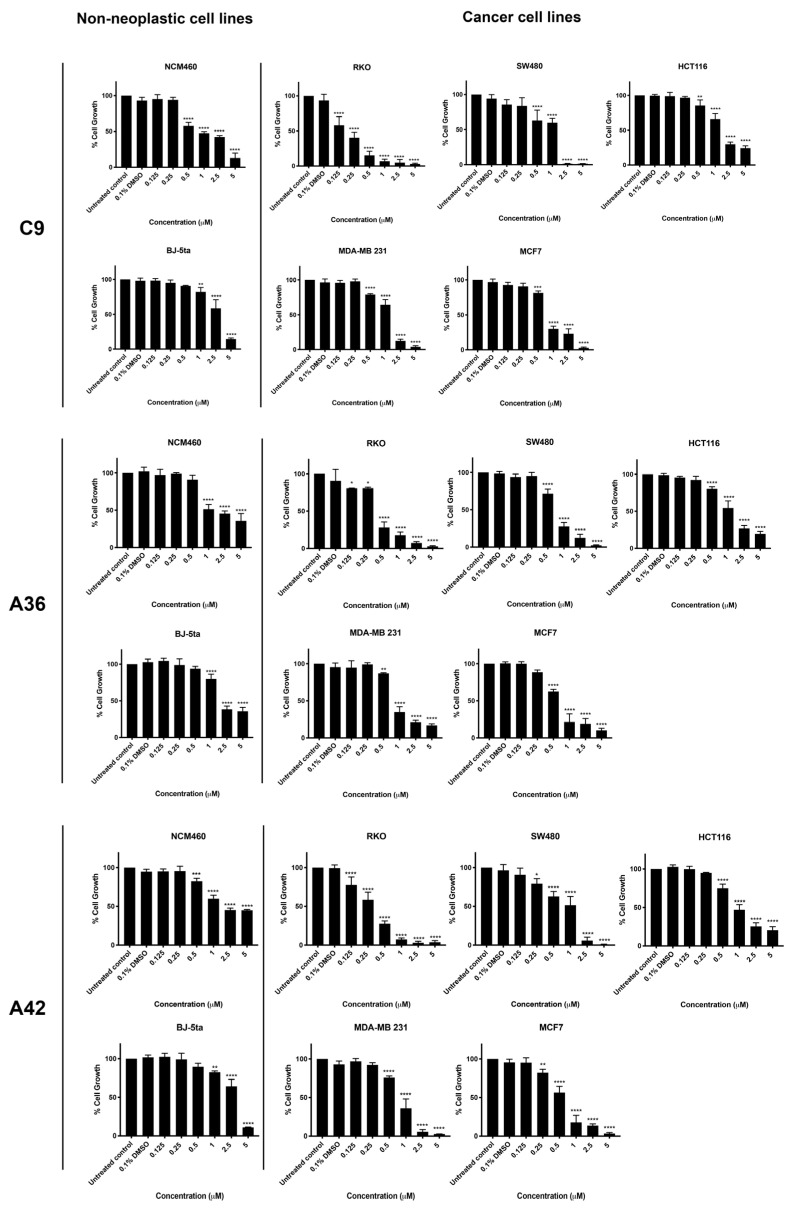
Effects of **C9**, **A36**, and **A42** on cell viability of BJ-5ta, NCM460, SW480, HCT116, RKO, MCF7, and MDA-MB-231 cell lines. Cell lines were subjected to increasing concentrations of compounds (0.125 μM, 0.25 μM, 0.5 μM, 1 μM, 2.5 μM, and 5 μM) or DMSO (0.1%) for 48 h. After incubation, cell viability was determined using the sulforhodamine B assay. The values are means with SD (*n* ≥ 3). The statistical analysis was conducted using two-way ANOVA. * *p* < 0.05, ** *p* < 0.01, *** *p* < 0.001, and **** *p* < 0.0001.

**Figure 3 cells-13-01385-f003:**
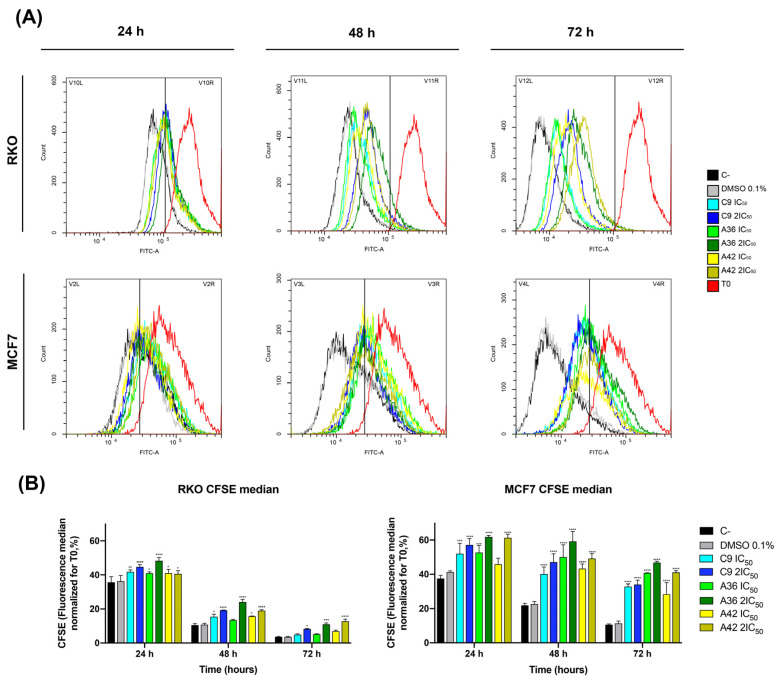
Effects of **C9**, **A36**, and **A42** on cell proliferation of RKO and MCF7 cell lines. (**A**) Representative histograms of RKO and MCF7 cell proliferation assays using carboxyfluorescein diacetate succinimidyl ester (CFSE). Cell lines were exposed to compounds **C9**, **A36**, and **A42** at IC_50_ and 2 × IC_50_ or DMSO (0.1%) for 0 h, 24 h, 48 h, and 72 h. (**B**) Quantification of the CFSE fluorescence median, values normalized to T0 after 24 h, 48 h, and 72 h of exposure. The values are means with SD (n ≥ 3). The statistical analysis was conducted using two-way ANOVA. * *p* < 0.05, ** *p* < 0.01, *** *p* < 0.001, and **** *p* < 0.0001.

**Figure 4 cells-13-01385-f004:**
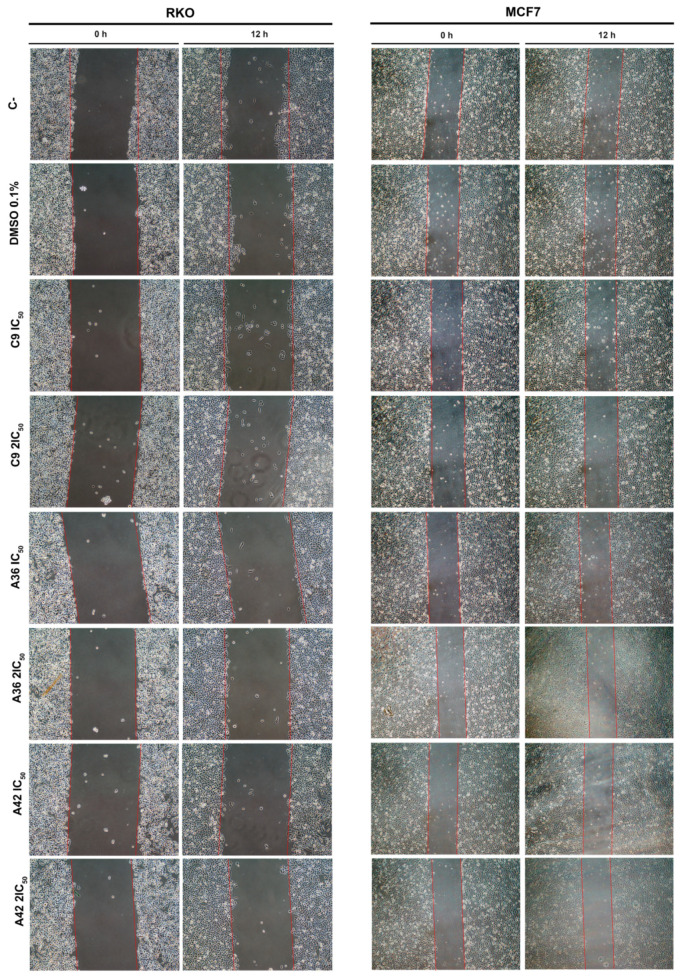
Effects of **C9**, **A36**, and **A42** on cell migration of RKO and MCF7 cell lines. Representative images (10× magnification) of wound healing assay after exposure of RKO and MCF7 cells to **C9**, **A36**, and **A42** at IC_50_ and 2 × IC_50_ for 0 h and 12 h. Untreated cells (C– and DMSO (0.1%)-exposed cells were used as negative controls. Quantification of the % of wound closure after 12 h. The values are means with SD (n ≥ 3). The statistical analysis was conducted using one-way ANOVA. * *p* < 0.05 and ** *p* < 0.01.

**Figure 5 cells-13-01385-f005:**
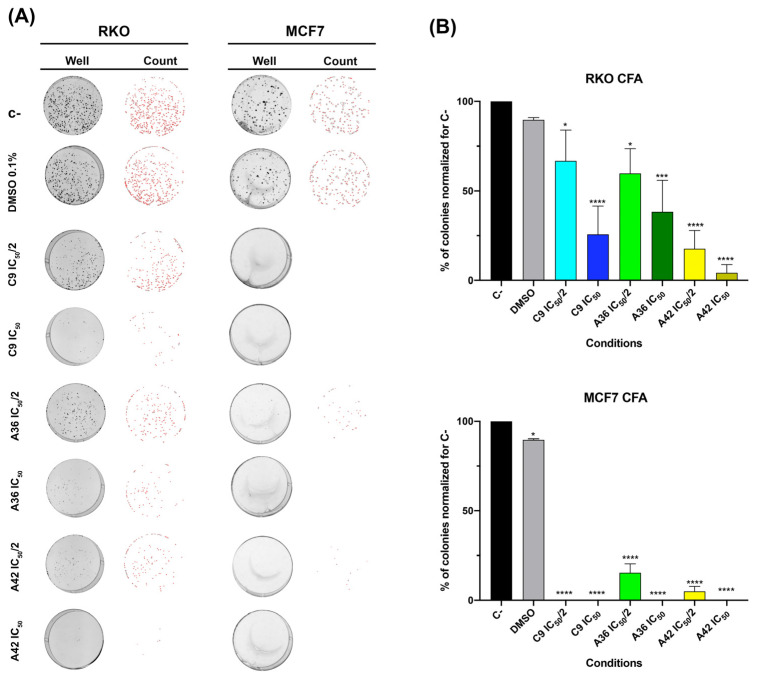
Effects of **C9**, **A36**, and **A42** on clonogenic ability of RKO and MCF7 cell lines. (**A**) Representative images (acquired with 5× magnification) of colony formation assay after RKO and MCF7 cells exposure to **C9**, **A36**, and **A42** at IC_50_/2 and IC_50_ for 48 h. Untreated cells (C–) and DMSO (0.1%)-exposed cells were used as negative controls. (**B**) Quantification of the number of colonies, values normalized to C–. The values are means with SD (n ≥ 3). The statistical analysis was conducted using one-way ANOVA. * *p* < 0.05, *** *p* < 0.001, and **** *p* < 0.0001.

**Figure 6 cells-13-01385-f006:**
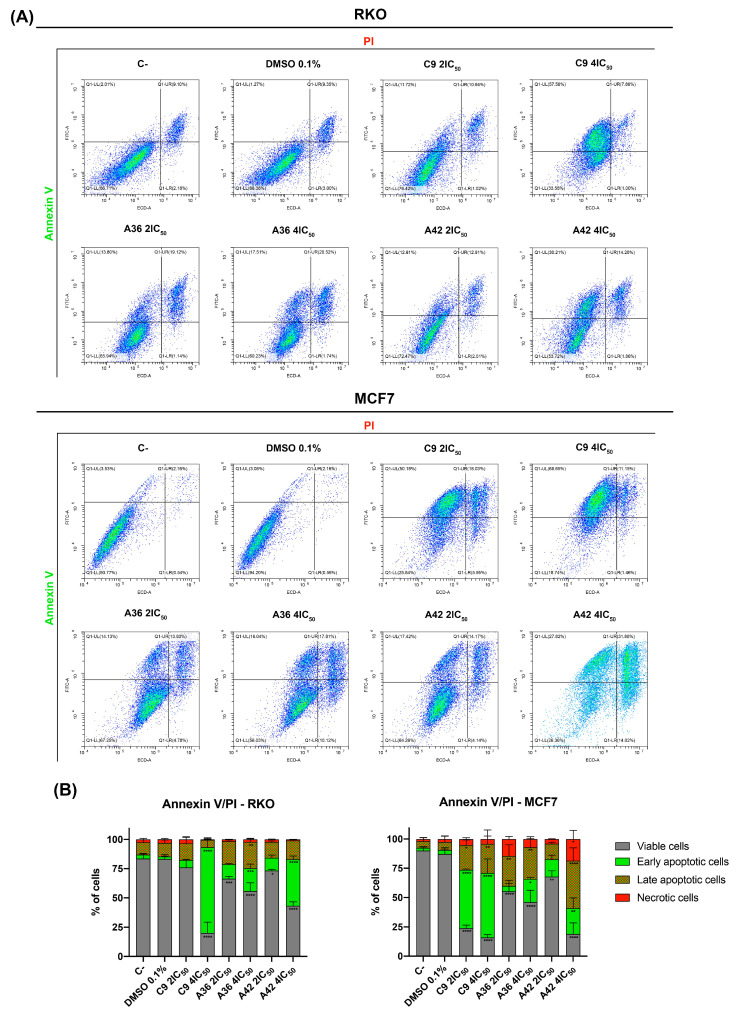
Effects of **C9**, **A36**, and **A42** on RKO and MCF7 cell death. (**A**) Representative bi-parametric dot plots of RKO and MCF7 Annexin V/PI analysis with Annexin V and propidium iodide (PI). Cell lines were exposed to compounds **C9**, **A36**, and **A42** at 2 × IC_50_ and 4 × IC_50_ or DMSO (0.1%) for 48 h. (**B**) Quantification of the cells in the different stages (viable, early apoptotic, late apoptotic, and necrotic). The values are means with SD (n ≥ 3). The statistical analysis was conducted using two-way ANOVA. * *p* < 0.05, ** *p* < 0.01, *** *p* < 0.001, and **** *p* < 0.0001.

**Figure 7 cells-13-01385-f007:**
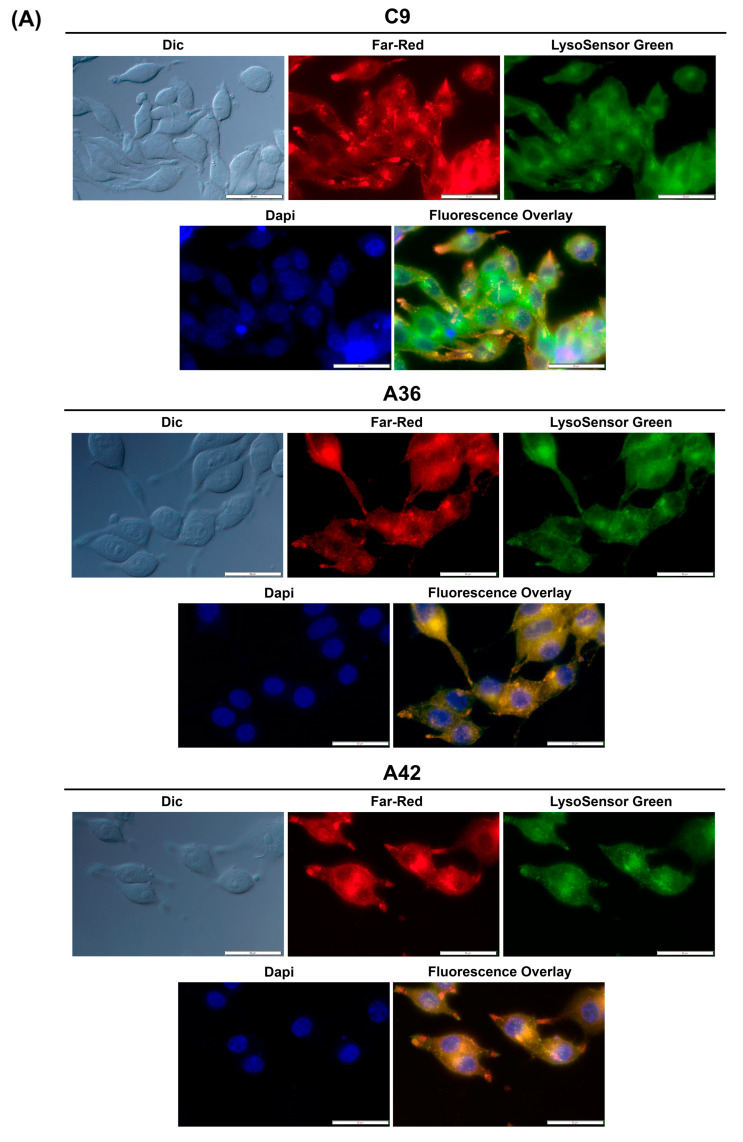
**C9**, **A36**, and **A42** lysosome accumulation. (**A**) Fluorescence microscopy images of RKO cells after incubation with **C9**, **A36**, and **A42** at IC_50_, cells were co-stained with 2 μM of LysoSensor Green and co-stained with 10 μg/mL of DAPI. (**B**) Fluorescence microscopy images of MCF7 cells after incubation with **C9**, **A36**, and **A42** at IC_50_, cells were co-stained 2 μM of LysoSensor Green and co-stained with 10 μg/mL of DAPI. Scale bar 50 μm.

**Figure 8 cells-13-01385-f008:**
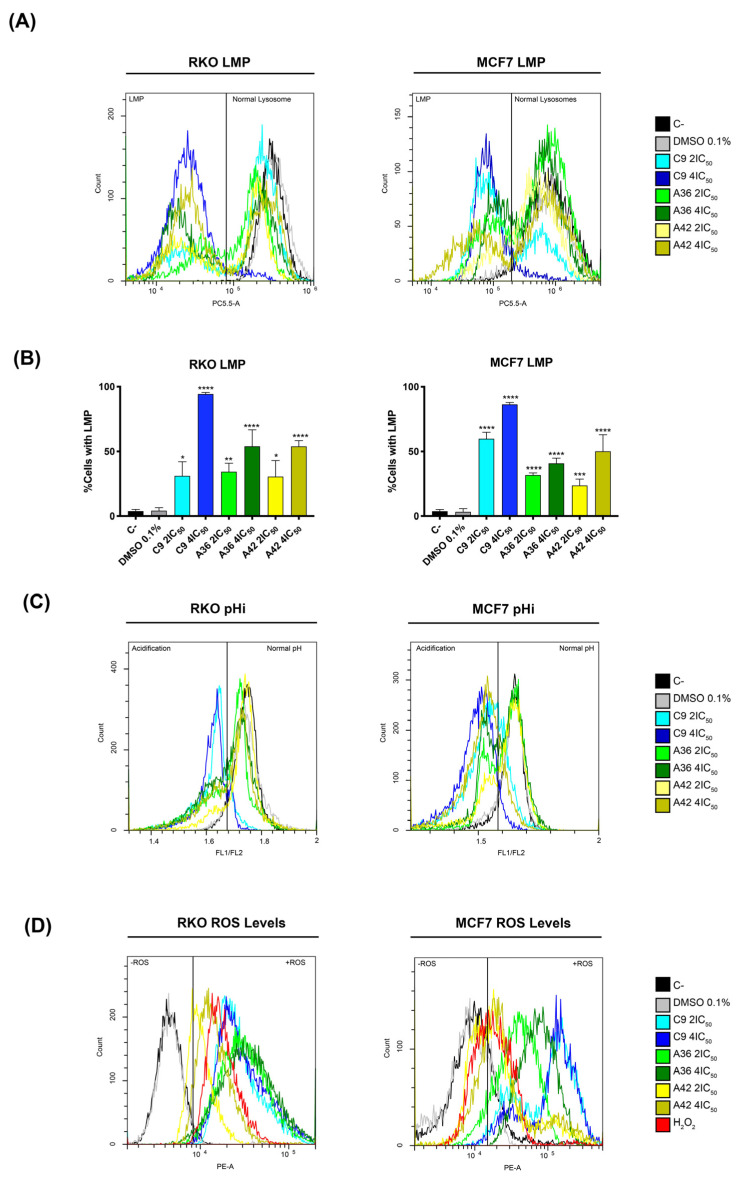
Effects of **C9**, **A36**, and **A42** on LMP, pHi, and ROS levels. Cell lines were exposed to compounds **C9**, **A36**, and **A42** at 2 × IC_50_ and 4 × IC_50_ or DMSO (0.1%) for 48 h. (**A**) Representative histograms of RKO and MCF7 LMP analysis with acridine orange (AO). (**B**) Quantification of the cells in LMP induction. The values are means with SD (n ≥ 3). The statistical analysis was conducted using two-way ANOVA. * *p* < 0.05, ** *p* < 0.01, *** *p* < 0.001, and **** *p* < 0.0001. (**C**) Representative histograms of RKO and MCF7 intracellular acidification with BCECF-AM, indicated by the fluorescence ratio between BCECF-AM green (FITC-FL1) and red fluorescence (PE-FL2) (FL1/FL2). (**D**) Representative histograms of RKO and MCF7 ROS levels with DHE.

**Table 1 cells-13-01385-t001:** IC_50_ (μM) ± SD for compounds BaP1 (lead compound), **C9**, **A36**, and **A42** in BJ-5ta, NCM460, SW480, HCT116, RKO, MCF7, and MDA-MB-231 cell lines.

Cell Lines		IC_50_ (μM) ^a^ ± SD ^b^			Selectivity Index ^c^	
	BaP1 ^d^	C9	A36	A42	Cell Line IC_50_/BJ-5ta IC_50_	CRC IC_50_/NCM460 IC_50_
BJ-5ta ^e^	1.83 ± 0.06	2.55 ± 0.15	2.44 ± 0.18	2.58 ± 0.19		
NCM460 ^e^	12.80 ± 2.05	1.12 ± 0.13	2.06 ± 0.25	2.59 ± 0.31		
SW480 ^f^	5.60 ± 0.19	0.78 ± 0.09	0.72 ± 0.03	0.78 ± 0.07	0.33/3.27/3.38/3.31	2.28/1.44/2.86/3.32
HCT116 ^f^	1.90 ± 0.09	1.64 ± 0.09	1.28 ± 0.06	1.13 ± 0.07	0.97/1.55/1.91/2.25	6.73/0.68/1.61/2.29
RKO ^f^	1.40 ± 0.08	0.17 ± 0.01	0.37 ± 0.03	0.29 ± 0.01	1.30/15/6.59/8.9	9.14/6.59/5.56/8.93
MCF7 ^g^	0.92 ± 0.07	0.84 ± 0.06	0.63 ± 0.05	0.54 ± 0.02	1.9/3.04/3.86/4.80	
MDA-MB-231 ^g^	1.73 ± 0.03	1.19 ± 0.05	1.01 ± 0.09	0.79 ± 0.03	1.05/2.14/2.41/3.26	

^a^—Data are presented as means values; compounds were tested in triplicate. ^b^—Standard deviation. ^c^—Selectivity index of **BaP1**/**C9**/**A36**/**A42**, respectively. ^d^—Lead compound. ^e^—Non-neoplastic cell line. ^f^—CRC cell line. ^g^—Breast cancer cell line.

## Data Availability

The data presented in this study are available on request from the corresponding author.

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
