# Peer review of "Anticancer Activity of Benzo[a]phenoxazine Compounds Promoting Lysosomal Dysfunction"

_cells, 2024, doi:10.3390/cells13161385_

Round 1

Reviewer 1 Report

Comments and Suggestions for Authors

The paper builds on previous work disclosing compound BaP1 as a new anticancer compound. Here, three benzophenoxazine compounds (A9, A36, A42) are studied with respect to cancer cell line activity and mechanistic feature relating to lysosomal membrane permeabilization. The work is carried out and written to a good scientific standard and will be of interest to the journal readership. 

It is notable however that the selectivity indices between colorectal and breast cancer cell lines, and normal controls, are quite small. This might suggest some off-target toxicities when testing in more advanced (in vivo) cancer models, and is a point that the authors should address in their discussion. The authors should also include the chemical structure of their positive control BaP1 for comparative purposes.

Comments on the Quality of English Language

Good standard of written English throughout.

Author Response

Comment 1: The paper builds on previous work disclosing compound BaP1 as a new anticancer compound. Here, three benzophenoxazine compounds (A9, A36, A42) are studied with respect to cancer cell line activity and mechanistic feature relating to lysosomal membrane permeabilization. The work is carried out and written to a good scientific standard and will be of interest to the journal readership. It is notable however that the selectivity indices between colorectal and breast cancer cell lines, and normal controls, are quite small. This might suggest some off-target toxicities when testing in more advanced (in vivo) cancer models, and is a point that the authors should address in their discussion. The authors should also include the chemical structure of their positive control BaP1 for comparative purposes.

Response 1: Thank you for your comments. Indeed the compounds have a quite strong effect that is also noticeable in the non-neoplastic cell lines. Still, these cells are the most resistant to the compounds and have the highest IC50 values. Nevertheless, we agree with your observation regarding the possible off-target toxicities that might occur. This is something that we will only know once we proceed with the characterization and test the compounds in more complex models. Since C9, A36 and A42 had a lower IC50 values and higher selectivity indices than our lead compound BaP1 we are now planning the next steps of characterization that will include to perform in vivo tests for the 3 compounds with advanced cancer models where we expect to evaluate the off-target toxicities.

We included BaP1 chemical structure as we agree that is an advantage for comparative purposes.

Reviewer 2 Report

Comments and Suggestions for Authors

The article: Anticancer Activity of Novel Benzo[a]phenoxazine Compounds 2 Promoting Lysosomal Dysfunction

General comments

The goal of the work was the investigation of the cytological background for understanding the effects basis for phenoxazine derivatives with potent antitumor activity. This study was focused by lysosomal accumulation and their mechanism of action, particularly those governing cell cycle progression and apoptosis. 

The abstract is complete and with good data obtained and described in the manuscript.

The experimental part of the study was well designed and had proper analytical tools and data analysis. The results obtained are very promising and with the possibility of evaluated the anti-carcinogenic activities.

I suggest to the authors to expand the introduction to the manuscript with the supplementary details about tested compounds C9, A36, A42 and BaP1 (lead compound).

Author Response

Comment 1: I suggest to the authors to expand the introduction to the manuscript with the supplementary details about tested compounds C9, A36, A42 and BaP1 (lead compound).

Response 1: Thank you for your comment, we agree with your suggestion, as such we completed the introduction with some more information as highlighted on the manuscript. line 54 to 68. “Our research team has been focused on synthesizing new benzo[a]phenoxazine derivatives, evaluating their pharmacological activity, and elucidating their mechanisms of action in order to assess their potential applications. To classify the most potent synthetized compounds we apply a comparative approach involving the evaluation of their biological activity by determining their MIC (Minimum Inhibitory Concentration) using a standard microdilution method for activity testing with the yeast S. cerevisiae as a eukaryotic cell model [17,24–38]. We further evaluated one of our compounds, BaP1 (lead compound displaying a MIC of 25 μM) against colorectal cancer cells and discovered that it had promising anticancer activity. We discovered that this effect was directly related to its lysosomal accumulation and lysosomal membrane permeabilization [39]. In the present work, we evaluate the anticancer potential of the most potent compounds from our library C9, A36, and A42 with MICs of 6.25 μM, 1.56 μM and 0.78 μM respectively(Fig. 1) [31,40–43]. Our aim was to uncover their anticancer activity not only against colorectal cancer but also test their potential against breast cancer cells as well as to evaluate their intracellular targets.”

Reviewer 3 Report

Comments and Suggestions for Authors

João Carlos Canossa Ferreira and his team have reported on the “Anticancer Activity of Novel Benzo[a]phenoxazine Compounds Promoting Lysosomal Dysfunction.” They synthesized three benzo[a]phenoxazine derivatives and tested their anticancer activity against the RKO colorectal cancer cell line and MCF7 breast cancer cell line. In both cases, these compounds reduced cell proliferation, survival, and migration, accumulated in the lysosomes, and induced cell death accompanied by lysosomal membrane permeabilization (LMP), which increased intracellular pH and ROS accumulation. After reviewing this manuscript, it is evident that major revisions are needed before it can be published in the Cells journal.

1.     The authors have made a statement in lines 46-47 but did not provide a citation. Proper references should be included in this statement.

2.     The structure of the lead compound BaP1 should be included to help readers understand its composition.

3.     The authors must justify the use of the same control figure (RKO) in Figure 4 that was used in their previous manuscript for BaP1.

4.     These are not novel compounds; similar compounds have been reported by the same authors in previous works, which are cited in the manuscript. It appears the authors are trying to generate two manuscripts from one scheme.

5.     These compounds exhibit significant cytotoxicity (around 1.12-2.59 μM) against non-cancerous cell lines such as NCM460 and BJ-210 5ta. What potential applications could these properties suggest for the compounds?

6.     The C9 and A42 H NMR spectra do not match their structures.

7.     The structures and 1H NMR spectra are overlapping in the supporting information provided.

8.     The authors should explain which solvent or impurity is responsible for the peaks at 4.59 and 2.37 ppm.

9.     There are different ppm values for compound A36 in this manuscript compared to the previously published paper in Tetrahedron Letters 57 (2016) 3936–3941. This discrepancy needs to be addressed.

10.  The authors should provide HRMS data for the compounds.

Comments on the Quality of English Language

Minor editing of English language required

Author Response

Comment 1. The authors have made a statement in lines 46-47 but did not provide a citation. Proper references should be included in this statement.

Response 1: Thank you for your comment, we agree with your suggestion, as such we added 3 citations that sustain the statement.

Comment 2.     The structure of the lead compound BaP1 should be included to help readers understand its composition.

Response 2: Thank you for your comment, we agree with your suggestion, BaP1 structure was added to figure 1.

Comment 4.     These are not novel compounds; similar compounds have been reported by the same authors in previous works, which are cited in the manuscript. It appears the authors are trying to generate two manuscripts from one scheme.

Response 4: Thank you for your comment; the compounds described in this paper are unique, have specific functionalizations, and were designed and synthesized by our group, which is why we refer to them as Novel Compounds. Their synthesis, the study of their photophysical properties and the determination of antifungal activity were indeed reported in other publications as referred on the manuscript. Here our objective was to further characterize them and uncover their anticancer activity something that was never reported for C9, A36 and A42. However, to avoid inappropriate interpretation, we eliminated the word "novel" in the title and also throughout the manuscript. As a result the title becomes: “Anticancer Activity of Benzo[a]phenoxazine  Compounds Promoting Lysosomal Dysfunction.”

Comment 5.     These compounds exhibit significant cytotoxicity (around 1.12-2.59 μM) against non-cancerous cell lines such as NCM460 and BJ-210 5ta. What potential applications could these properties suggest for the compounds?

Response 5: Thank you for your comment; we agree with your observation; the compounds are very potent, causing some cytotoxicity in our control cell lines; however, we were able to obtain good selectivity indexes for the majority of cancer cells with the majority of the compounds. We are now moving with the characterization of these four compounds (C9, A36, A42 and BaP1), and the next step will be the evaluation of their possible off-target toxicities. This will involve performing in vivo tests with advanced cancer models, where we expect to further validate their targeting anticancer activity.

Comment 6. The C9 and A42 H NMR spectra do not match their structures.

Response 6: Thank you for your comment; the structure of C9 had an inaccuracy, but has now been corrected in Figure 1, so that the 1H and 13C NMR spectra are in agreement with it. The 1H NMR spectrum of compound C9 was revised. In case of A42, the spectrum presents peaks related to probably a trace amount of the nitrosated precursor with chemical shifts coinciding or very close to those resulting from benzophenoxazine, essentially in the aliphatic zone. The pattern of signals of the aromatic zone shows that it is benzophenoxazine A42.

Comment 7. The structures and 1H NMR spectra are overlapping in the supporting information provided.

Response 7: Thank you for your comment; the supporting information was improved, the overlap has been fixed.

 Comment 8.  The authors should explain which solvent or impurity is responsible for the peaks at 4.59 and 2.37 ppm.

Response 8: In the 1H NMR spectrum of compound C9, now shown, the signal at 2.37 no longer exists. Regarding the signal at 4.59, it may be related to some residual solvent, but its presence is not significant.

Comment 9.There are different ppm values for compound A36 in this manuscript compared to the previously published paper in Tetrahedron Letters 57 (2016) 3936–3941. This discrepancy needs to be addressed.

Response 9: The differences recorded in the chemical shift values are around 0.1- 0.2 ppm, which may be related to the solvents from which the solid was obtained.

Comment 10.  The authors should provide HRMS data for the compounds.

Response 10: As mentioned in the manuscript, the complete synthesis and characterization of the three compounds, including HRMS results, has been previously reported. In the present work, the syntheses were repeated to obtain more amount of the compounds to allow the studies that were carried out. In these situations, it is usual to confirm the structures of the compounds by 1H NMR spectroscopy and comparing the spectra with those previously obtained. Therefore, this was the procedure followed.

Round 2

Reviewer 3 Report

Comments and Suggestions for Authors

 The authors have revised the manuscript and addressed my previous comments. I believe it has been significantly improved and is now suitable for publication in the Cells Journal.

Comments on the Quality of English Language

Minor editing of English language required